# A goat experimental epistaxis model: Hemostatic effect of stop nosebleeds device

Amani Abu-Shaheen [1]*, Shroaq Saleh Aljanobui[2], Falah Hassan Almohanna[3], Mohsen Ayyash [4], Sumayyia Marar[1], Goran Matic[3], Mohammed Hazazi[3], Juneil Batalla[3], Maaweya Awadalla [1], Muaawia A. Hamza[1]

1 Department of Scientific Writing and Biostatistics, Research Center, King Fahad Medical City, Riyadh Second Health Cluster, Riyadh, Kingdom of Saudi Arabia, 2 Inventor, Riyadh, Saudi Arabia, 3 Department of Comparative Medicine, King Faisal Specialist Hospital and Research Centre, Riyadh, Saudi Arabia, 4 School of Mathematical Sciences, Universiti Sains Malaysia, USM, Penang, Malaysia

* aabushaheen@kfmc.med.sa

## Abstract

To stop epistaxis, a Saudi medical invention called "Stop Nosebleeds" with a patent was created. It is a silicone-based, adjustable device that is applied externally to the bridge of the nose and speeds up clotting by decreasing the temperature and the compressing effect of the device on the nose This study aims to examine the efficacy of the device in attaining hemostasis in goats with anterior nasal bleeding. An animal experimental nasal-bleeding model was conducted on ten goats in collaboration between King Fahad Medical City and King Faisal Specialist Hospital and Research Center, Riyadh, Saudi Arabia. The left nostril was used as interventional wounds and the right nostril was used as control. Control wounds were treated with manual compression for 20 minutes. Interventional wounds were treated with the Stop Nosebleeds device. The second group of animals was heparinized and 2 wounds were created in each animal, one of which was being treated with the Stop Nosebleeds device, while the other was being treated with manual compression. The mean bleeding duration in the experimental group (both heparinized and non-heparinized goats) was 45.5 seconds (SD = 8.2), compared to 206.0 seconds (SD = 75.7) in the control group (U = 0.00, p < 0.001). The device stopped bleeding in 52.0 seconds and 39.0 seconds in the heparinized and non-heparinized experimental groups. While, the manual compression of epistaxis stopped bleeding in 252.5 seconds (SD = 84.7) and 159.2 seconds (SD = 15.7) in the heparinized and non-heparinized control groups, respectively. The device shows promise for clinical application, however, further research with larger sample sizes and human clinical trials are needed to validate its efficacy and safety. Additionally, the use of a goat model presents limitations in directly translating these results to human clinical practice, and these differences should be careful.

**Data availability statement:** All relevant data are within the manuscript and its Supporting Information files.

**Funding:** Grant management department, Research Center, King Fahad Medical City, Riyadh, Saudi Arabia.

**Competing interests:** The authors have declared that no competing interests exist.

## Introduction

Epistaxis refers to the active bleeding from inside the nose or nasal cavity [1]. It is one of the most prevalent otolaryngology emergencies, with nearly 60% of people in the United States experiencing it and 10% needing medical attention. Children (2–10 years) and adults (50–80 years) are most commonly affected [2].

Epistaxis is classified as anterior or posterior with over 90% of the cases originating from the anterior nasal septum which is vascularized by the Kisselbach's plexus while posterior bleeds account for 10% of epistaxis cases [3]. Nose bleeding can result from a variety of local and systemic factors. Local causes include trauma, neoplasia, septal abnormality, iatrogenic causes, and inflammatory diseases, while systemic causes commonly include hypertension, age, bleeding diathesis, and alcoholism [3].

Treatment strategies for epistaxis include pressing the nostrils, putting a cold compress on the bridge of the nose, hot water irrigation, anterior nasal packing, cauterization, hemostatic or vasoconstriction medicines, and cryotherapy [3–5]. However, these approaches can increase the risk of infection, septal perforation, and aspiration. Therefore, research is ongoing to develop new methods for managing epistaxis [6,7].

The 2020 "Clinical Practice Guideline: Nosebleed (Epistaxis)" was developed to guide healthcare practitioners on aspects related to nosebleeds. It includes recommendations for first-line treatment of active bleeding, with a compression of the lower third of the nose for at least 5 minutes or longer shown to be highly effective [8].

A Saudi medical device "Stop Nosebleeds" was developed to manage epistaxis. The device is classified as a medical device by the Saudi Food and Drug Authorization and nominated by the Saudi Ministry of Health to participate in the 50th International Geneva Invention Exhibition 2025 at Palexpo, Geneva. This patented device does not interfere with the components of the natural coagulation process, making it potentially useful in individuals with both normal hemostatic parameters and those with primary or secondary hemostatic disorders. Moreover, the device can serve as a first-aid tool providing a temporary solution until the patient reaches the emergency room. Various epistaxis management models have been evaluated in numerous studies [9–11]. However, the effectiveness and clinical efficacy of different management models still need further investigation [12]. In comparison to other existing hemostatic devices such as anterior nasal packing, which are associated with complications like pain [13], the Stop Nosebleeds device is not painful, easy to use, and does not require prophylactic antibiotics and analgesics.

While this device is ultimately designed for human use, goats provide a suitable animal model for studying epistaxis because of their anatomical and histological similarities to humans [14]. Additionally, due to ethical concerns surrounding the use of other animals such as cats and dogs in experimental research, goats have become a popular alternative [14]. Thus, this study aims to examine the efficacy of the Stop Nosebleeds device on attaining hemostasis in goats with anterior nasal bleeding.

## Materials and methods

An animal experimental nasal-bleeding model was conducted using the Stop Nosebleeds device (Fig 1) in collaboration between the Research Center, King Fahad Medical City, Riyadh, Saudi Arabia, and the Laboratory Animal Service, Department of Comparative Medicine, King Faisal Specialist Hospital and Research Center, Riyadh, Saudi Arabia.

### Stop nosebleeds device

The device is an external, silicon-based, adjustable tool designed to fit various nose shapes and is applied externally over the nasal bridge (Fig 2). The device increases the speed of blood clotting in the nasal area by lowering the temperature of the exterior walls, which is induced by a chemical reaction inside the device's cavity. The device works by rubbing the ends of the piece together by hand, causing the water bag in the device's chamber to rupture, triggering a chemical reaction with ammonium nitrate crystals, which leads to a rapid drop in the temperature. As a result of this mechanism of action, the outer walls of the device reach four degrees Celsius and inflate within 45 seconds. This cooling effect, combined with the compressive action of the device on the nose, speeds up blood clotting by both reducing the temperature and applying pressure. Previous studies have shown that cooling and compression can contribute to hemostasis by inducing clot formation [4,15,16].

Ten male local goats were used in this study. The goat's age was between 6 and 12 months. Goats with any disease or abnormalities were excluded from this study. The animals were purchased locally and kept in a room at a constant temperature (22°C±4°C) with a 12-hour light/dark cycle, 55±5% humidity, normal diet, and free access to tap water several days before the investigation and were fasted overnight before any procedures. The study protocol, housing, and care for animals followed the animal care and use committee at the Department of Comparative Medicine King Faisal Specialized Hospital and Research Center, Riyadh, Saudi Arabia.

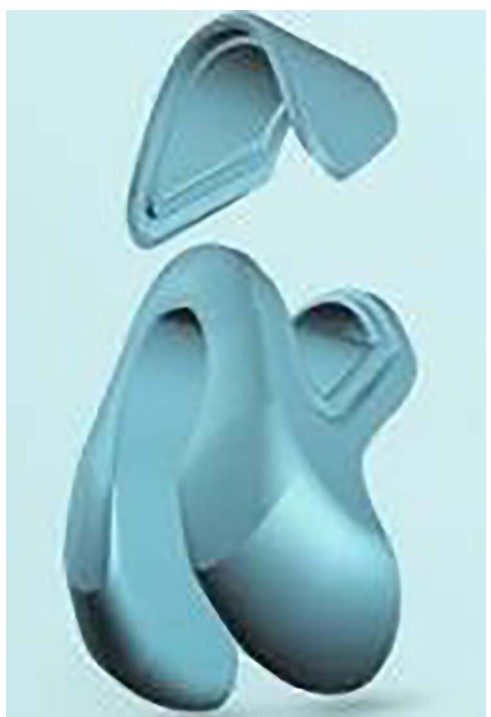

**Fig 1. Stop Nosebleeds device used to stop epistaxis.**

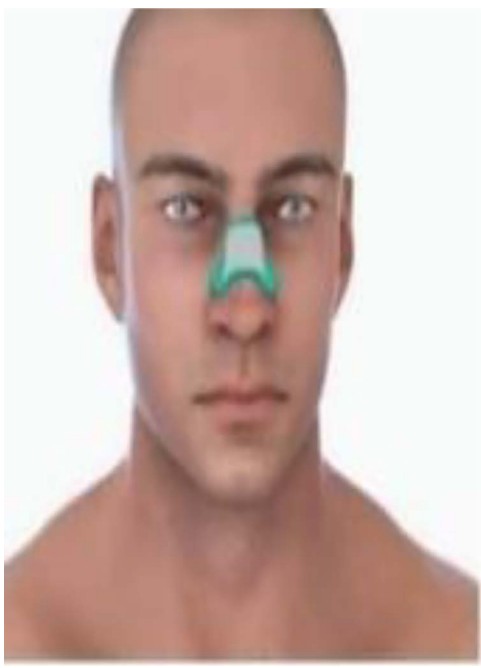

**Fig 2. Stop the Nosebleeds device by applying pressure and cooling over the nasal bridge.**

## Study protocol

The study was conducted at the Laboratory Animal Service, Department of Comparative Medicine King Faisal Specialist Hospital and Research Center, Riyadh, Saudi Arabia. Ten adult goats, aged 6–12 months with an average weight of 26.44 kg (SD = 2.72 kg), were included in this study, such that the left nostril was used as the experimental group, and the right nostril was used as the control (Fig 3). All animals were indigenous Saudi goats of the Harri breed (species: Capra hircus), sourced from a reputable supplier where they were born and raised in a controlled and healthy environment. Prior to the experiment, routine blood work, including complete blood count (CBC) and chemistry tests, was conducted, and the animals were clinically observed for any signs of abnormal bleeding or clotting, such as excessive bruising, prolonged bleeding from minor cuts, or spontaneous bleeding.

The animals were allowed to acclimatize for one week before anesthesia. They were fasted for 18 hours and water was withheld for 2 hours before the induction of anesthesia. Flunixin (1.1 mg/kg) was administered 30 minutes before anesthesia induction. Ketamine (10 mg/kg) and xylazine (0.2 mg/kg) were administered via intramuscular injection, followed by atropine (0.15–0.3 mg/kg) via the same route. An intravenous catheter was then placed in the jugular vein. Anesthesia was induced with propofol (4 mg/kg) and then maintained with sevoflurane (2–3%). The animal was intubated and mechanically ventilated, and a stomach tube was placed to reduce the risk of aspiration. The depth of anesthesia was assessed by monitoring eye position, muscle tone, and jaw tone. Throughout the procedure, heart rate, respiratory rate, body temperature, and oxygen saturation were monitored. Blood pressure was not monitored during the procedure.

Standardized, full-thickness, mucosal wounds were created on the anterior nasal septum of the goats using a 3-mm surgical punch. A caliper was used to ensure consistency in the size and depth of all punch defects. Additionally, the veterinary surgeon performing the punch defects was highly trained to apply consistent technique and pressure, with each procedure, ensuring uniformity in both size and depth. The surgical punch was applied to the nasal mucosa of the right and left anterior nasal septum using slight pressure. The punch was rotated 90° clockwise and 90° counterclockwise to

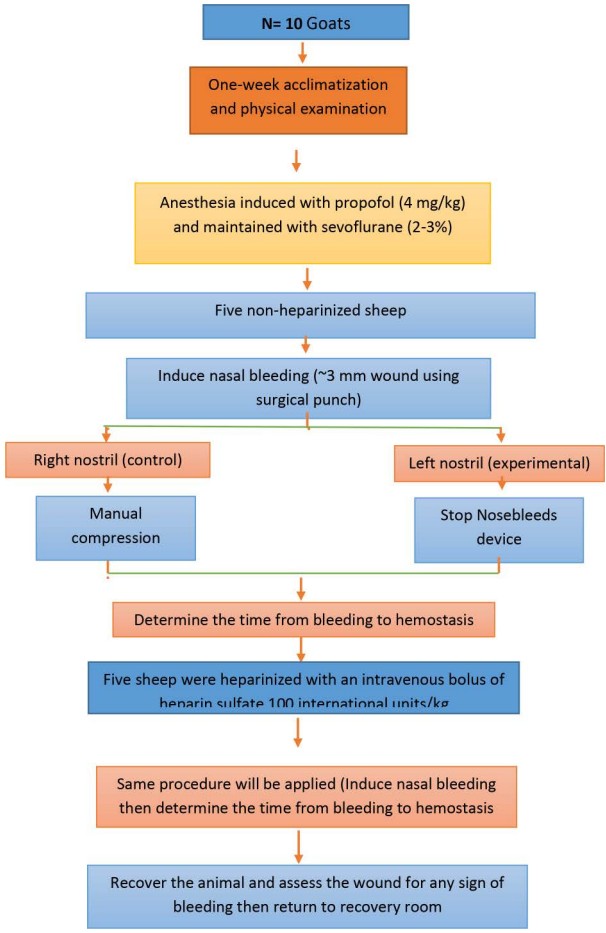

**Fig 3. Roadmap of the study protocol and procedures.**

yield a full-thickness mucosal cut without harming the septal cartilage. This resulted in bleeding within 10–20 seconds of injury in all cases.

Five goats were heparinized (with an intravenous bolus of heparin sulfate 100 international units/kg) and went through the same anesthesia protocol. Two wounds were then created in each animal, one of which was treated with the Stop Nosebleeds device, while the other was treated with manual compression. Only the surgical punches resulted in consistent and uniform wounds that bled for at least 30 seconds in the unheparinized animal. Manual compression was always performed first on the right nostril (control group) to ensure consistency in the procedure, with control wounds treated according to routine practice by applying manual compression to the cartilaginous part of the nose for 20 minutes. The time to hemostasis for these control wounds was then recorded. The left nostril (experimental group) was treated afterward using the Stop Nosebleed device. This sequential approach allowed for a comparison of the device's effectiveness with standard manual compression while maintaining a controlled experimental setup.

To minimize the potential influence of local hemostasis from the first intervention on the second, we ensured that an adequate time interval was maintained between procedures. Additionally, both nostrils were treated as independent sites since each had its vascular supply, reducing the likelihood of crossover effects. For both, manual compression and the device, the pressure was gradually released to check for ongoing bleeding. Additionally, a second indication for

hemostasis was the clot formation. Once the clot was formed, the pressure was released or the device was removed. Afterward, the animal was monitored for any signs of recurrent bleeding such as pooling blood or oozing [17].

Randomization was not applied in this setup to maintain methodological consistency and to ensure that manual compression, as the standard treatment, was always performed first under controlled conditions. This approach enabled a clear comparison between interventions while minimizing variability from external factors that could arise from altering the order of treatments.

After completing the experiment, the animals were allowed to recover from anesthesia and then transferred to the post-operative care room. Vital signs (temperature, heart rate, respiratory rate, mucous membrane (gums) color, and capillary refill time) and animal activities/behaviors were assessed and documented in the post-operative recovery room by the veterinary technician. Nasal wounds were observed for 24 hours for any sign of re-bleeding. After the study's completion, the animals were returned to the comparative medicine department.

The data were collected using a predesigned data collection sheet that consisted of six sections: goat characteristics (including the age and gender), lab profile in relation to the timing of the examination pre-and post-intervention, procedures without heparin and with heparin for both intervention and control wounds, and observation of re-bleeding after 24 hours. The lab profile section specifically tracked the goat's physiological status, focusing on the assessment of re-bleeding. Baseline data were recorded before the intervention. After 24 hours, follow-up lab tests were conducted to detect any changes in the parameters and identify signs of re-bleeding.

## Outcomes

The primary outcome was the duration of bleeding, defined as the time elapsed from the initiation of bleeding (i.e., the creation of full-thickness wounds) until its cessation. The secondary outcome was the difference in bleeding duration between heparinized and non-heparinized subjects within each group (experimental and control). Hemostasis was measured from the moment of injury and was considered complete when no visual blood pooling was noted.

## Ethical consideration

This study was approved by the Institutional Animal Care and Use Committee (IACUC) of Comparative Medicine King Faisal Specialist Hospital and Research Center, Riyadh, Saudi Arabia (RAC# 2230013) in 2022. All methods were performed according to the ARRIVE (Animal Research: Reporting of In Vivo Experiments) and IACUC guidelines and regulations.

## Sample size estimate

The sample size was determined based on the methodology outlined by Festing et al. [18] based on power analysis and according to the guidelines reported by Charan and Kantharia [19], ensuring the study was adequately powered to detect meaningful differences. Assuming a standard deviation of 1.25 minutes in bleeding time for each goat and aiming to detect a 2-minute mean difference between the experimental and control groups, the effect size (D) was calculated to be 1.6 (D = difference in means/ standard deviation). Therefore, with a significance level of 5%, a power of 90%, a two-sided test, and an effect size of 1.6, the required sample size is 10 goats [18]. The mean difference was defined based on a similar study [20].

## Statistical analysis

Study variables were described using means and standard deviations (SDs). To compare bleeding duration or bleeding time between the experimental and control groups, as well as between heparinized and non-heparinized goats, the Mann-Whitney U test was used. Statistical significance was defined at p-values less than 0.05. Data analyses were conducted using the Statistical Package for the Social Sciences (SPSS), version 26.0.

## Results

Ten adult goats were included in this study. Table 1 presents the bleeding times measured for wounds on the right (control) and left (experimental) nostrils. The mean bleeding duration in the experimental group (both heparinized and non-heparinized goats) was 45.5 seconds compared to 206.0 seconds in the control group.

The result of Levene's test for equality of variances was found to be statistically significant, indicating a significant difference in variability between the experimental and control groups (F = 6.343, p = 0.021). The control group exhibits greater variability (SD = 75.7) than the experimental group (SD = 8.2), indicating that the hemostatic intervention reduced variability in bleeding times, as illustrated in Fig 4.

The results of the Mann-Whitney test indicate that the mean bleeding time in wounds treated with the Stop Bleeding device (intervention) was significantly shorter than in the control group (U = 0.00, p < 0.001). During the 24-hour observation period, none of the wounds experienced recurrent bleeding.

The Stop Nose Bleeding Device stopped bleeding in 52.0 seconds (SD = 5.4) and 39.0 seconds (SD = 2.6) in the heparinized and non-heparinized experimental groups, respectively, as reported in Table 2. The difference in average bleeding times between the heparinized and non-heparinized interventional groups was statistically significant (p = 0.004). In contrast, manual compression took longer to stop bleeding, with a duration of 252.5 seconds (SD = 84.7) and 159.2 seconds (SD = 15.7) in the heparinized and non-heparinized control groups, respectively, with the difference also being statistically significant (p = 0.004), as illustrated in Fig 5.

As for postoperative care, all goats received three doses of antibiotic (Amoxykel, 15 mg/kg via intramuscular route) and analgesic (Ketovet, 50 mg/kg via intramuscular route) over three days (one dose per day) and were monitored for potential wound re-bleeding.

**Table 1. Bleeding times in goats with epistaxis on the right and left nostril.**

| Goat ID | Left Nostril (Experimental Group) | Right Nostril (Control Group) |
|---|---|---|
| 1 | 38 | 161 |
| 2 | 36 | 182 |
| 3 | 43 | 159 |
| 4 | 38 | 138 |
| 5 | 40 | 156 |
| 6 H | 47 | 190 |
| 7 H | 53 | 212 |
| 8 H | 45 | 216 |
| 9 H | 55 | 400 |
| 10 H | 60 | 246 |
| Mean (SD) | 45.5 (8.2) | 206.0 (75.7) |
| Median (Range) | 44.0 (24) | 186.0 (262) |
| Levene's test, F (p-value) | 6.343 (0.021) | |
| Mean difference (SE) | − 160.5 (24.07) | |
| Mann-Whitney Test Statistic (U) | 0.00 | |
| P-value | < 0.001 | |

H: Heparinized; SD: Standard deviation; SE: Standard error; Mean difference = Experimental − Control.

Time is presented in seconds.

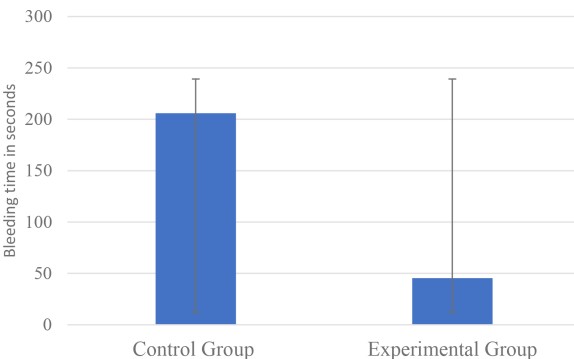

**Fig 4. Average bleeding time in both groups.**

**Table 2. Mean bleeding duration in heparinized and no heparinized goats.**

|  | Left Nostril | Right Nostril |
|---|---|---|
| Non-Heparinized (n=5) | 39.0 (2.6) | 159.2 (15.7) |
| Heparinized (n=5) | 52.0 (5.4) | 252.8 (84.7) |
| P-value (Mann-Whitney Test) | 0.004 | 0.004 |

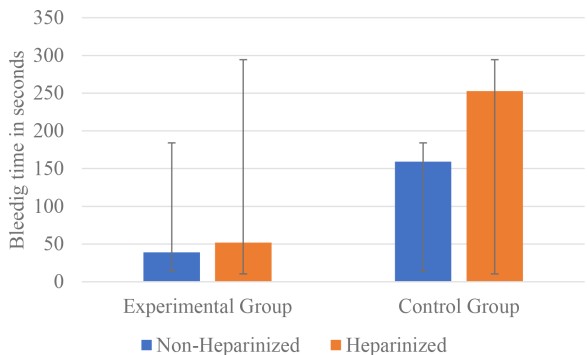

**Fig 5. Mean bleeding duration between the non-heparinized and heparinized goats for both the experimental and control groups.**

## Discussion

The study findings suggest that the Stop Nosebleeds device has a significant hemostatic effect on epistaxis, and was found to be effective in shortening the duration of bleeding compared to manual compression in a goat epistaxis model.

In the present study, the mean bleeding duration in the experimental group was 45.5±8.2 seconds, compared to 206.0±75.7 seconds in the control group. This finding aligns with previous research highlighting the effectiveness of various hemostatic devices and agents in reducing bleeding times in animal models compared to control groups. For instance, Kurtaran et al. [20] observed a significantly shorter mean bleeding time of 98±17 seconds in a rabbit epistaxis model treated with Ankaferd BloodStopper (ABS) to 266±36 seconds in the control group. Similarly, Yurttaş et al. [21] compared the hemostatic effects of ABS and microporous polysaccharide hemospheres (MPH), reporting that the MPH group exhibited a shorter mean bleeding time of 11.5±2.1 seconds, compared to ABS (20.5±2.9 seconds) and the control (41.3±10.8 seconds). Additionally, Kelleş et al. [22] suggested that the topical nasal application of ABS controls

epistaxis more quickly at 30 seconds than gelatin foam (90 seconds), adrenaline + lidocaine (90 seconds), and control (210 seconds) in a rabbit bleeding model. Moreover, our findings are also consistent with those of Singer et al. [23], who demonstrated that oxidized cellulose achieved faster hemostasis in a standardized porcine epistaxis model as compared to those who clotted on their own (92 seconds vs 209 seconds, P < 0.001). Collectively, these studies indicate that various hemostatic agents, including tacrolimus, oxidized cellulose, ABS, MPH, and OCA, show promise in effectively managing epistaxis. These agents generally outperform traditional methods like nasal packing, providing faster and more reliable hemostasis. Therefore, the current study provides additional evidence of the effectiveness of the Stop Nosebleeds Device in reducing the duration of bleeding in a goat model compared to traditional manual compression procedures.

The higher variability in bleeding times observed in the control group can be attributed to natural physiological differences in coagulation and the absence of hemostatic intervention. Studies have shown that bleeding time is influenced by several factors such as platelet function, vascular integrity, and external factors like injury severity [24]. Hemostatic interventions, such as medical devices or topical agents, significantly reduce variability by standardizing clotting mechanisms and minimizing extreme outliers [25]. Additionally, variations in coagulation markers and natural hemostatic responses have been well-documented, reinforcing the role of biological variability in untreated cases [26]. This supports the effectiveness of the stop-nosebleed device in reducing both bleeding duration and variability in the goat model.

Similar to our study findings, previous studies indicated the higher efficacy in devices using compression to stop epistaxis; for instance, the swimmer's nose clip method, which maintains constant pressure on the bleeding site, along with ice packs, has also shown promise in improving outcomes [27]. Another study found that nose clips were more effective than manual compression in controlling severe epistaxis [6].

Nevertheless, the observed variability in bleeding times within the control group (SD = 75.7 seconds) in the current study raises important considerations and could reflect a range of factors that may have contributed to inconsistency including variations in the wound creation process or differences in the individual responses of the animals. To address this variability, future studies conducted with a larger sample size may help to better account for the natural variability in animal responses.

No re-bleeding incidents were observed during the 24-hour monitoring period, indicating that the hemostasis achieved with both methods was stable and effective, consistent with other animal epistaxis models [20,28]. The marked decrease in bleeding time suggests that the Stop Nosebleeds device may be highly effective for managing epistaxis, however, since epistaxis can result from various causes and conditions, the device's effectiveness may vary depending on the underlying etiology. Achieving rapid hemostasis is crucial in both emergency and routine care to prevent complications from prolonged bleeding. This device could minimize the necessity for more invasive procedures and potentially enhance patient outcomes. Moreover, in real-world clinical settings, the Stop Nosebleeds device, with its ease of use and quick action, can serve as an effective first-aid solution. Its compact size and ease of storage make it a highly portable option. For patients on anticoagulants, it may be considered safe as it is pain-free and eliminates the need for analgesics and antibiotics.

It offers significant advantages for first-aid use, providing a quick and easy solution in emergencies; portability, with a compact design making it suitable for public or home settings; and safety in anticoagulated patients, offering a safer alternative to traditional methods that may exacerbate bleeding risks.

However, this study is subject to some limitations, including the small sample size and the lack of long-term follow-up to study the effects of the device and ensure sustained hemostasis and safety. Moreover, the difference in the anatomy of a goat's muzzle from that of the human nose and other anatomical variations must be considered while interpreting the study findings.

While the findings are robust, future studies on humans are needed to confirm the current results. Moreover, it would be beneficial to investigate the long-term effects of the Stop Nosebleeds device to ensure sustained hemostasis and safety. Future research should explore the comparative effectiveness of the Stop Nosebleeds device against other established hemostatic agents and devices. This could include head-to-head trials in various clinical and animal models to better

understand its relative advantages. Investigating the mechanisms by which the device achieves hemostasis could also provide valuable insights and potentially lead to further improvements.

Our findings are expected to serve as the foundation for future randomized controlled trials on humans, shedding light on the need for those patients to have a friendly use device as a first-aid tool to provide a temporary solution until the patient arrives at the emergency room.

## Conclusion

This study shows that the Stop Nosebleeds device significantly reduces bleeding time in a goat model of epistaxis, demonstrating its potential as an effective hemostatic intervention. These results are supported by similar findings in previous studies involving other hemostatic devices and animal models While these results are promising, further research is required to validate the device's efficacy and safety in humans. Larger sample sizes, human clinical trials, and comparative studies with other established hemostatic devices are necessary to confirm the device's clinical utility. Moreover, the long-term effects of pressure and cold from the device on the nasal mucosal integrity should be studied.

## Supporting information

**S1 File. Data sheet.**
(PDF)

## Acknowledgments

We are grateful to the Research Center, King Fahad Medical City, Riyadh Second Health Cluster, Riyadh, Saudi Arabia.

## Author contributions

**Conceptualization:** Amani Abu-Shaheen, Shroaq Saleh Aljanobui, Falah Hassan Almohanna, Mohsen Ayyash, Sumayyia Marar, Goran Matic, Mohammed Hazazi, Maaweya Awadalla, Muaawia A. Hamza.

**Data curation:** Amani Abu-Shaheen, Falah Hassan Almohanna, Goran Matic, Mohammed Hazazi, Juneil Batalla.

**Formal analysis:** Mohsen Ayyash.

**Funding acquisition:** Amani Abu-Shaheen.

**Investigation:** Amani Abu-Shaheen, Sumayyia Marar.

**Methodology:** Amani Abu-Shaheen, Falah Hassan Almohanna, Mohsen Ayyash, Sumayyia Marar, Goran Matic, Juneil Batalla, Maaweya Awadalla, Muaawia A. Hamza.

**Project administration:** Amani Abu-Shaheen, Sumayyia Marar, Mohammed Hazazi, Muaawia A. Hamza.

**Resources:** Amani Abu-Shaheen, Shroaq Saleh Aljanobui, Mohammed Hazazi.

**Supervision:** Amani Abu-Shaheen.

**Validation:** Amani Abu-Shaheen, Maaweya Awadalla.

**Visualization:** Muaawia A. Hamza.

**Writing – original draft:** Amani Abu-Shaheen, Shroaq Saleh Aljanobui, Falah Hassan Almohanna, Mohsen Ayyash, Sumayyia Marar, Goran Matic, Mohammed Hazazi, Juneil Batalla, Maaweya Awadalla, Muaawia A. Hamza.

**Writing – review & editing:** Amani Abu-Shaheen, Shroaq Saleh Aljanobui, Falah Hassan Almohanna, Mohsen Ayyash, Sumayyia Marar, Goran Matic, Mohammed Hazazi, Juneil Batalla, Maaweya Awadalla, Muaawia A. Hamza.

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
