## [Decision Letter · Decision Letter 0]

7 Jan 2025

Dear Dr. Abu-Shaheen,

Thank you for submitting your manuscript to PLOS ONE. After careful consideration, we feel that it has merit but does not fully meet PLOS ONE’s publication criteria as it currently stands. Therefore, we invite you to submit a revised version of the manuscript that addresses the points raised during the review process.

We look forward to receiving your revised manuscript.

Kind regards,

Mehmet Baysal

Academic Editor

PLOS ONE

“We are grateful to the King Fahad Medical City Research Center, Riyadh, Saudi Arabia, for providing us with the financial support to complete this project (IRF# 022-036).  “

4. In the online submission form you indicate that your data is not available for proprietary reasons and have provided a contact point for accessing this data. Please note that your current contact point is a co-author on this manuscript. According to our Data Policy, the contact point must not be an author on the manuscript and must be an institutional contact, ideally not an individual. Please revise your data statement to a non-author institutional point of contact, such as a data access or ethics committee, and send this to us via return email. Please also include contact information for the third party organization, and please include the full citation of where the data can be found.

Reviewers' comments:

Reviewer's Responses to Questions

**Comments to the Author**

1. Is the manuscript technically sound, and do the data support the conclusions?

Reviewer #1: Yes

Reviewer #2: Yes

Reviewer #3: Partly

Reviewer #4: No

Reviewer #5: Yes

Reviewer #6: Yes

Reviewer #7: Partly

Reviewer #8: Partly

2. Has the statistical analysis been performed appropriately and rigorously?

Reviewer #1: Yes

Reviewer #2: Yes

Reviewer #3: N/A

Reviewer #4: Yes

Reviewer #5: Yes

Reviewer #6: Yes

Reviewer #7: Yes

Reviewer #8: I Don't Know

3. Have the authors made all data underlying the findings in their manuscript fully available?

Reviewer #1: Yes

Reviewer #2: Yes

Reviewer #3: No

Reviewer #4: No

Reviewer #5: Yes

Reviewer #6: Yes

Reviewer #7: Yes

Reviewer #8: Yes

4. Is the manuscript presented in an intelligible fashion and written in standard English?

Reviewer #1: Yes

Reviewer #2: No

Reviewer #3: No

Reviewer #4: Yes

Reviewer #5: Yes

Reviewer #6: Yes

Reviewer #7: No

Reviewer #8: No

Reviewer #1: The idea is fair new. Please check the document with track changes.-------------------------------------------------------------------------------------------------------------------------------------------------------------------------------------------------------------------------------------------------------------------------------------------------------------------------------------------------------

Reviewer #2: The manuscript titled "A Goat Experimental Epistaxis Model: Hemostatic Effect of Stop Nosebleeds Device" addresses a clinically relevant issue by testing a novel hemostatic device. The study design and results are promising; however, there are areas where the manuscript could be improved.

Abstract:

The abstract effectively summarizes the study but would benefit from specifying areas requiring further research, such as long-term outcomes or human clinical trials. Additionally, it should explicitly acknowledge the inherent limitation of translating findings from a goat model to human clinical applications to provide a more balanced perspective.

Introduction:

1.While the introduction highlights the importance of managing epistaxis and introduces the device, it lacks a robust justification for selecting goats as the model organism. Including a comparison to other animal models in the literature would strengthen this section.

2.The novelty of the "Stop Nosebleeds" device is mentioned, but it would be helpful to include a brief comparison with existing hemostatic devices or agents.

Material and methods:

1.Device Mechanism: The cooling mechanism's contribution to hemostasis is not directly validated. Adding data on nasal tissue temperature or citing similar studies could substantiate this claim.

2.Bleeding Time Measurement: The criteria for determining complete hemostasis (absence of visible blood pooling) should be supported with additional references or validation.

Results:

1.The variability in bleeding times (e.g., control group SD = 75.7 seconds) raises questions about potential inconsistencies in wound creation or animal responses. Addressing this variability in the discussion would be useful.

2.The results rely solely on text and tables, which, while clear, lack complementary visual aids like graphs or charts.

Discussion:

1.Indirect comparisons: The discussion summarizes the findings of other hemostatic agents but does not directly compare their outcomes (e.g., bleeding times, efficacy rates) to the Stop Nosebleeds device. It lacks quantitative comparisons, such as how the bleeding times achieved with ABS or MPH differ from those achieved with the Stop Nosebleeds device.

2.No focus on device uniqueness: While it mentions similarities to other devices, the unique advantages or disadvantages of the Stop Nosebleeds device (e.g., simplicity, cooling mechanism) compared to ABS, MPH, or oxidized cellulose are not highlighted.

3.Contextualize for clinical use: Address why the Stop Nosebleeds device could be advantageous in real-world scenarios (e.g., first-aid use, portability, or safety in anticoagulated patients).

Reviewer #3: Dear authors,

Thank you for this interesting manuscript about a new device to stop nasal bleeding. Although I think this manuscript may be interesting as initial research to keep developing the device, I found there are major thing that need to be addressed before publication.

General

English revision necessary

No preoperative blood sampling? How do you know coagulation was normal? And platelet count or other variables that may affect bleeding? Although each animal was its own control, it may have affected results, please, state this in the text.

Major changes

Pg 7, lines 7-8. How did you ensure all punch defects were same depth and affecting same structures? It can massively change bleeding. Please, state this in the text.

Pg 7, line 15. How did you control time to haemostasis? Relieving the pressure periodically? How was this performed (both with manual compression and the device)? Please, state this in the text.

Pg7, line 7. After induction, how was anaesthetia monitored?? How do you know arterial blood pressure and the other variables were normal and they did not affect bleeding? How did you control anaesthetic depth? Any preoperative analgesia administered? Please, state this in the text.

Pg 9. Isoflurane or sevoflurane were used? Change it in the text or figure as necessary.

Pg 7, line 13. Manual compression was always performed first? How was controlled that local haemostasia was not affecting second bleeding?? Why not randomizing? Please, state this in the text.

Pg 10, line 2. Your primary outcome was the number of wounds which stop bleeding. There is no information about this in the results section. If that was your primary outcome, why the sample size calculation was performed with time (following ARRIVE guidelines)? Please, state this in the text.

Pg 10, line 10. In the sample size calculation, you obtained 10 animals, but then, you performed 2 different experiments with only 5 animals per group, explain that (can we trust results of the two separate groups? Enough number of animals per group?). Where did you obtain your mean difference between groups to perform the sample size calculation? Previous studies? Preliminary study? Please, state this in the text.

Pg 12, results. Redundant information, no need for text and table telling the same.

Pg 13, line 11. Antibiotic and anti-inflammatory administration should be moved to Materials & Methods.

Pg 14, lines 4 - 6. How is your device compared to other methods? Advantages and disadvantages?

Pg 14, lines 7 - 9. Paragraph about antibiotics / anti-inflammatory not necessary.

Pg 14, lines 9 - 11. Haemostasis was achieved with both methods, please, state this in the text to make it clear.

Pg 14, lines 12 - 13. You cannot state that it may be highly effective in clinical settings, because, as you stated in the introduction, epistaxis can be caused by multiple situations and pathologies and you just performed a punch (lesion) please, rephrase and change this in the text to make it clear.

Pg 14, discussion. How do we know that the pressure and cold from the device did not affect nasal mucosal integrity long-term? Any studies on that? Please, state this in the text.

Thank you very much for this interesting manuscript.

Your faithfully.

Reviewer #4: El tema es bastante interesante pero debe mejorar lo siguiente, el documento no se encuentra en el formato de la revista ni tampoco cumple con los parámetros mínimos de ella. Ojo con el número de línea. El resumen se excede las 300 palabras. No hay homogeneidad en todo el documento. Se encuentra muy desordenado. No se encuentra el título de materiales y métodos, como a la vez la guía para el protocolo. Las figuras se deben poner “Fig.1”, “Fig.2” etc. Las tablas no están en el formato que corresponde. Falta coherencia en el escrito. Solo el 20.6% son citas del 2019 al 2024 significa que el 79.3% son fuentes menores de los 5 años. No se encuentran las citas número 26, 27 y 28 nombradas en la bibliografía. Todas las citas no se encuentran con las normas Vancouver que es lo que indica la revista si no una mezcla de esta con normas apa. Tener cuidado, y también tengan en cuenta la homogeneidad en estas mismas.

Reviewer #5: Abstract: Appropriate and crysp

Introduction: Appropriate and satisfactoryReview

Materials & Methods: Research techniques and methodology used are appropriate with standard statistical design application. Planning and conduct of experiments are appropriate as per the topic chosen.

Results & Appendices: The result have been appropriately presented in an elaborate and systemic manner, precisely interpreted and properly discussed with appropriate discussion.

Discussion: Discussion is critical with valid justification addressing all the objectives mentioned

Summary: Appropriate and satisfactory

Reviewer #6: - Heparin used in control group and other with stop bleeding device and regular

- ways of conducting bleeding in experimental animal not clear

- Statistical analysis can be modified with positive and control group with standard deviation

- IACUC year agreement of protocol approval should be stated for experimental design

- References need more up date from 2019 until now

Reviewer #7: In the abstract section, the method and results should have been given piece by piece, and should have been written without specifying them separately. The photographs used in the material method section should be referenced if they are the photographs taken by the authors themselves. There are font differences in the article. If the right and left nasal mucosa of the same animal were to be used, the number of animals should have been higher, it is quite insufficient. Clinical parameters indicating the health status of the animals should have been clearly presented. The specified device is available for use in humans, Animal and human nasal mucosa and epithelium are histologically different. This difference may cause problems in comparing the results with other articles. Human structure and tissue are closer to laboratory animals, so why was a goat chosen, it would have been more appropriate to do it on a laboratory animal.The reference articles used are old, the number of articles in the last 5 years is low.

Reviewer #8: In the abstract and a large part of the manuscript it's not clear if the tested device is expected ultimately to be used in human or goat nasal bleeding. In the abstract, it might be wise to briefly outline the mechanism of action of this device, the descrition of the method and the chronology of the acts are not clear, the conclusion of this study on goats seems to have been extrapolated to human without sufficient caution in the terms used.

In the first part of the introduction it's not clear if the authors talk about goats or humans.

In the material and method section, the description of the device is not clear. An expansion of the device is mentionned but this is not described precisely enough. Even though it seems obvious, there is one or two references missing on the effect on temperature ans compression on the speed of clot formation and stopping of bleeding. A reference would also be useful to support the fact stated here that goat's nasal blood flow most closely resembles that of human beings. On the other part the anatomy of goat's muzzle seems a little bit different from these of the human's nose, so the good coaptation of the device in either species might be questionable. What is normal diet of the experimental animals during the study.

Some informations about the experimental animals mentioned in the result part like age and weight would have a better place in the material and method section. Description of anesthesy doesn't mention the use of premedication (which could influence the vasomotor status of the mucosa and so the bleeding). Is this the case ? The description and chronology of experimental acts as the road map of the protocol is not clear. The lab profil mentioned as part of a data collection sheet should be described with more detail as the protocol of assessment of "re-bleeding" (timing of examination and criteria)

In table 1 goats which were heparinized should be identified.

In the context of One Health and good practices about antibiotic use is it really necessary to administer antibiotics for two 5mm diameter wounds surgicaly performed in the cranial part of the nasal mucosa. At any rate, this precision should be taken up in the material and methods rather then in the results section. Again in the discussion there is confusion between the condition, the results ant the conclusions in goats and humans. The other studies in animal models mentioned used mainly topical or general chemical agents to stop bleeding and seem to me difficult to compare with or to support the findings in the reviewed study about a mechanical device. In the other hand, if we come back to the idea of carrying out this experiment on goats, we can see that most of the other animals models mainly concern rabbits.

**Do you want your identity to be public for this peer review?** For information about this choice, including consent withdrawal, please see our Privacy Policy

Reviewer #1: **Yes: ** Prof. Dr. Qaes Talb Al-Obaidi

Reviewer #2: No

Reviewer #3: No

Reviewer #4: No

Reviewer #5: No

Reviewer #6: **Yes: ** Amira Saad Helal Hassenin

Reviewer #7: No

Reviewer #8: **Yes: ** Caudron Isabelle

---

## [Author Response · Author response to Decision Letter 1]

19 Mar 2025

18 March, 2025

The Editor

PLOS ONE

Subject: Submission of the first revision of Manuscript: PONE-D-24-52201

Dear Editor,

Thank you very much for the evaluation and consideration of our manuscript titled “A Goat Experimental Epistaxis Model: Hemostatic Effect of Stop Nosebleeds Device”, submitted for exclusive consideration for publication with PLOS One.

We greatly appreciate the insightful comments and feedback provided by the external referees. We have made revisions throughout the manuscript in response to the comments and included an itemized response below. All changes in the manuscript have been noted in track changes.

We believe that the modifications have strengthened the manuscript and successfully prepared for publication.

We appreciate the time and effort spent in this peer review process and thank you for your continued consideration of our work.

Please contact us if you need any further revisions or information.

Authors

Response to Reviewer Comments

Reviewer #1:

The idea is fair new. Please check the document with track changes.

Response: Thank you for the comments. All suggested edits have been incorporated in the revised manuscript.

Reviewer #2:

The manuscript titled "A Goat Experimental Epistaxis Model: Hemostatic Effect of Stop Nosebleeds Device" addresses a clinically relevant issue by testing a novel hemostatic device. The study design and results are promising; however, there are areas where the manuscript could be improved.

Abstract:

The abstract effectively summarizes the study but would benefit from specifying areas requiring further research, such as long-term outcomes or human clinical trials. Additionally, it should explicitly acknowledge the inherent limitation of translating findings from a goat model to human clinical applications to provide a more balanced perspective.

Response: We appreciate the valuable feedback. The suggested points have been included in the abstract of the revised manuscript.

Introduction:

1.While the introduction highlights the importance of managing epistaxis and introduces the device, it lacks a robust justification for selecting goats as the model organism. Including a comparison to other animal models in the literature would strengthen this section.

Response: Thank you for your suggestion. We have added a few lines in the introduction that justify the selection of goats in this study in comparison to other animal models.

2.The novelty of the "Stop Nosebleeds" device is mentioned, but it would be helpful to include a brief comparison with existing hemostatic devices or agents.

Response: Thank you for the comment. In the revised manuscript we have included a brief comparison of Stop Nosebleeds device with traditional hemostatic devices such as anterior nasal packaging.

Material and methods:

1.Device Mechanism: The cooling mechanism's contribution to hemostasis is not directly validated. Adding data on nasal tissue temperature or citing similar studies could substantiate this claim.

Response: Thank you for the valuable comment. In the revised manuscript, we have cited a study that supports the effect of cooling in achieving hemostasis.

2.Bleeding Time Measurement: The criteria for determining complete hemostasis (absence of visible blood pooling) should be supported with additional references or validation.

Response: Thank you for the valuable comment. In the revised manuscript, we have supported the determination of hemostasis with a reference.

Results:

1.The variability in bleeding times (e.g., control group SD = 75.7 seconds) raises questions about potential inconsistencies in wound creation or animal responses. Addressing this variability in the discussion would be useful. (insert)

Response: We appreciate the insightful comment. Accordingly, we have discussed the variability in bleeding times in the revised manuscript.

2.The results rely solely on text and tables, which, while clear, lack complementary visual aids like graphs or charts.

AUTHOR: Thank you for your suggestion. Accordingly, two graphs have been included in the revised manuscript.

Discussion:

1. Indirect comparisons: The discussion summarizes the findings of other hemostatic agents but does not directly compare their outcomes (e.g., bleeding times, efficacy rates) to the Stop Nosebleeds device. It lacks quantitative comparisons, such as how the bleeding times achieved with ABS or MPH differ from those achieved with the Stop Nosebleeds device.

RESPONSE: Thank you for the suggestion. We have edited the paragraph to include the specific bleeding times related to ABS and MPH reported in the studies that allow a quantitative comparison to the Stop Nosebleeds device.

2.No focus on device uniqueness: While it mentions similarities to other devices, the unique advantages or disadvantages of the Stop Nosebleeds device (e.g., simplicity, cooling mechanism) compared to ABS, MPH, or oxidized cellulose are not highlighted.

RESPONSE: Thank you for the suggestion. In the revised manuscript, we have included a paragraph on the uniqueness of the Stop Nosebleeds device.

3.Contextualize for clinical use: Address why the Stop Nosebleeds device could be advantageous in real-world scenarios (e.g., first-aid use, portability, or safety in anticoagulated patients).

RESPONSE: We appreciate the valuable suggestion. These points have been included in the revised manuscript.

Reviewer #3:

Dear authors,

Thank you for this interesting manuscript about a new device to stop nasal bleeding. Although I think this manuscript may be interesting as initial research to keep developing the device, I found there are major thing that need to be addressed before publication.

General

English revision necessary

Response: English language editing was carried out while revising the manuscript.

No preoperative blood sampling? How do you know coagulation was normal? And platelet count or other variables that may affect bleeding? Although each animal was its own control, it may have affected results, please, state this in the text.

Response: The animals were sourced from a reputable supplier, where they were born and raised in a controlled, healthy environment. Routine blood work was conducted prior to the experiment including CBC and chemistry tests and they were clinically observed for any signs of abnormal bleeding or clotting, such as excessive bruising, prolonged bleeding from small cuts, or spontaneous bleeding. Epistaxis can be life threatening and the primary objective of this study was to evaluate the effectiveness of the Stop Nosebleeds device in controlling epistaxis, particularly in emergency situations, regardless of whether the patient has a coagulation disorder. Additionally, the device has been tested on heparinized animals, showing a significant reduction in bleeding duration compared to manual compression.

This information has been updated in the revised manuscript.

Major changes

Pg 7, lines 7-8. How did you ensure all punch defects were same depth and affecting same structures? It can massively change bleeding. Please, state this in the text.

RESPONSE: Thank you for the comment. The suggested information has been included in the revised manuscript as follows: A caliper was used to verify the consistency in size and depth of all punch defects. Additionally, the veterinary surgeon performing the punch defects is highly trained to apply the same technique and pressure with each procedure, ensuring uniformity in both size and depth.

Pg 7, line 15. How did you control time to haemostasis? Relieving the pressure periodically? How was this performed (both with manual compression and the device)? Please, state this in the text.

Response: Thank you for your valuable comments. We have clarified the above queries as follows in the revised manuscript:

Manual compression was always performed first on the right nostril (control group) to ensure consistency in the procedure. The left nostril (experimental group) was treated afterward using the Stop Nose Bleed Device. This sequential approach allowed us to compare the effectiveness of the device with standard manual compression while maintaining a controlled experimental setup. To minimize the potential influence of local hemostasis from the first intervention on the second, we ensured that an adequate time interval was maintained between procedures. Additionally, both nostrils were treated as independent sites since each had its own vascular supply, reducing the likelihood of crossover effects. Randomization was not applied in this setup to maintain methodological consistency and to ensure that manual compression, as the standard treatment, was always performed under controlled conditions first. This approach allowed for a clear comparison between interventions while preventing variability in external factors that could arise from alternating the order of treatments. For both, manual compression and the device, the pressure was gradually released to check for ongoing bleeding. Additionally, a second indication for hemostasis is the clot formation. When the clot is formed the pressure is released or the device is removed. After relieving pressure or removing the device, the animal is monitored for any signs of recurrent bleeding e.g. pooling blood or oozing.

Pg7, line 7. After induction, how was anaesthetia monitored?? How do you know arterial blood pressure and the other variables were normal and they did not affect bleeding? How did you control anaesthetic depth? Any preoperative analgesia administered? Please, state this in the text.

Response: After induction, a pulse oximetry was placed on the goat’s tongue to continuously measure the heart rate and the oxygen saturation of the blood. Additionally, physical indicators such as eye position, muscle tone, jaw tone, etc.; were used to assess the anesthesia depth. Blood pressure was not monitored during the procedure, only heart rate, body temperature, ECG and oxygen saturation and they were all normal throughout the procedure. Flunixin Melgumine 1.1 mg/kg was given pre-operatively via IV.

Pg 9. Isoflurane or sevoflurane were used? Change it in the text or figure as necessary.

Response: Sevoflurane was used. The change has been documented in the revised manuscript.

Pg 7, line 13. Manual compression was always performed first? How was controlled that local haemostasia was not affecting second bleeding?? Why not randomizing? Please, state this in the text.

Response: Manual compression was always performed first. The decision not to randomize the order was based on the goal of maintaining a consistent approach and controlling variables in a predictable and reproducible manner. This information has been included in the revised manuscript.

Pg 10, line 2. Your primary outcome was the number of wounds which stop bleeding. There is no information about this in the results section. If that was your primary outcome, why the sample size calculation was performed with time (following ARRIVE guidelines)? Please, state this in the text.

Response: We appreciate the reviewer comment. The primary and secondary outcomes are now clearly defined in the manuscript. Since the primary outcome is the bleeding time, so we estimated the sample size based on our primary outcome. This information has been included in the revised manuscript.

Pg 10, line 10. In the sample size calculation, you obtained 10 animals, but then, you performed 2 different experiments with only 5 animals per group, explain that (can we trust results of the two separate groups? Enough number of animals per group?). Where did you obtain your mean difference between groups to perform the sample size calculation? Previous studies? Preliminary study? Please, state this in the text.

Response: Thank you for your insightful comments. The procedure of sample size calculation is now clearly defined in the manuscript as requested. The sample size was determined based on the methodology outlined by Festing et al. (22), ensuring the study was adequately powered to detect meaningful differences and thus the number of animals per group was appropriate for statistical analysis. Regarding the mean difference used for the sample size calculation, it was defined based on a similar study conducted by Kurtaran et al. (1).

Pg 12, results. Redundant information, no need for text and table telling the same.

Response: We have edited this section to make it concise.

Pg 13, line 11. Antibiotic and anti-inflammatory administration should be moved to Materials & Methods.

Response: Thank you for the comment. As per the suggestion, this information has been removed from the discussion.

Pg 14, lines 4 - 6. How is your device compared to other methods? Advantages and disadvantages?

Response: Thank you for the suggestion. In the revised manuscript, we have discussed the unique advantages and disadvantages of the device compared to other methods.

Pg 14, lines 7 - 9. Paragraph about antibiotics / anti-inflammatory not necessary.

Response: Thank you for the comment. As per the suggestion, this information has been removed from the discussion.

Pg 14, lines 9 - 11. Haemostasis was achieved with both methods, please, state this in the text to make it clear.

Response: Thank you for the suggestion. We have clarified this in the revised manuscript.

Pg 14, lines 12 - 13. You cannot state that it may be highly effective in clinical settings, because, as you stated in the introduction, epistaxis can be caused by multiple situations and pathologies and you just performed a punch (lesion) please, rephrase and change this in the text to make it clear.

Response: Thank you for the comment. As per the reviewer’s suggestion, these lines have been rephrased in the revised manuscript.

Pg 14, discussion. How do we know that the pressure and cold from the device did not affect nasal mucosal integrity long-term? Any studies on that? Please, state this in the text.

Response: We have included this point as a suggestion for future studies.

Thank you very much for this interesting manuscript.

Your faithfully.

Reviewer #4:

The topic is quite interesting but the document is not in the format of the journal nor does it meet the minimum parameters of the journal.

Be careful with the line number. The abstract exceeds 300 words. There is no homogeneity throughout the document. It is very disorderly. The title of materials and methods is missing, as well as the guide for the protocol. The figures should be put “Fig.1”, “Fig.2” etc. The tables are not in the correct format. There is a lack of coherence in the writing. Only 20.6% are citations from 2019 to 2024 which means that 79.3% are sources under 5 years old. Citations number 26, 27 and 28 are not listed in the bibliography. All citations are not found with Vancouver standards, which is what the journal indicates, but a mixture of this with apa standards. Be careful, and also take into account the homogeneity in these same.

Response: Thank you for the comments. We have edited to abstract to meet the word limit of 300 words. Recent references have been included wherever possible. Other formatting suggestions have also been made.

Reviewer #5:

Abstract: Appropriate and crysp

Introduction: Appropriate and satisfactoryReview

Materials & Methods: Research techniques and methodology used are appropriate with standard statistical design application. Planning and conduct of experiments are appropriate as per the topic chosen.

Results & Appendices: The result have been appropriately presented in an elaborate and systemic manner, precisely interpreted and properly discussed with appropriate discussion.

Discussion: Discussion is critical with valid justification addressing all the objectives mentioned

Summary: Appropriate and satisfactory

Reviewer #6: -

Heparin used in control group and other with stop bleeding device and regular

- ways of conducting bleeding in experimental animal not clear

Response: Thank you for the comment. We have added more details regarding how bleeding was conducted in the experimental group.

- Statistical analysis can be modified with positive and control group with standard deviation

Response: W

---

## [Decision Letter · Decision Letter 1]

30 Apr 2025

A Goat Experimental Epistaxis Model: Hemostatic Effect of Stop Nosebleeds Device

PONE-D-24-52201R1

Dear Dr. Abu-Shaheen,

We’re pleased to inform you that your manuscript has been judged scientifically suitable for publication and will be formally accepted for publication once it meets all outstanding technical requirements.

Kind regards,

Mehmet Baysal

Academic Editor

PLOS ONE

Additional Editor Comments (optional):

Reviewers' comments:

Reviewer's Responses to Questions

**Comments to the Author**

Reviewer #1: All comments have been addressed

Reviewer #2: All comments have been addressed

Reviewer #4: (No Response)

Reviewer #5: All comments have been addressed

2. Is the manuscript technically sound, and do the data support the conclusions?

Reviewer #1: Yes

Reviewer #2: Yes

Reviewer #4: Partly

Reviewer #5: Yes

3. Has the statistical analysis been performed appropriately and rigorously?

Reviewer #1: Yes

Reviewer #2: Yes

Reviewer #4: No

Reviewer #5: Yes

4. Have the authors made all data underlying the findings in their manuscript fully available?

Reviewer #1: Yes

Reviewer #2: Yes

Reviewer #4: Yes

Reviewer #5: Yes

5. Is the manuscript presented in an intelligible fashion and written in standard English?

Reviewer #1: Yes

Reviewer #2: Yes

Reviewer #4: No

Reviewer #5: Yes

Reviewer #1: (No Response)

Reviewer #2: The author has addressed my concerns, particularly regarding quantitative comparisons, including bleeding times with ABS, MPH, and the Stop Nosebleeds device, as well as the device's uniqueness. These clarifications enhance the manuscript’s rigor and clarity.

I appreciate the revisions and have no further major concerns. The manuscript is now more robust and will contribute valuable insights to the field.

Reviewer #4: Al documento no se le realizó ningún cambio anteriormente sugerido, no está en el formato de la revista, no tiene línea de seguimiento continua, no está bien escrito en inglés, no hay coherencia en algunos párrafos. Introducción: "active bleeding from inside the nose or nasal cavity" - "active bleeding from the nasal cavity" (más conciso; "inside" y "nasal cavity" es redundante). Además, la gramática: “Children (2-10 years) and adults (50-80 years) are most commonly affected.” - es mejor y más natural decir: “...are the most commonly affected age groups.” También, “Treatment strategies for epistaxis include pressing the nostrils...” - Suena más informal para un artículo científico; sería mejor reemplazarlo por: “Initial management includes digital compression of the nostrils…"

Resultados: Mejorar la redacción, coherencia y conectores; también pueden usar términos como “recorded” o “summarizes”, “achieve hemostasis” y “substantially more time”, que son más apropiados científicamente. Además de reorganizar algunas frases largas para que fueran más fáciles de leer.

Discusión: Igualmente ocurre en la discusión, no hay coherencia ni claridad entre ideas, falta mejorar la precision del lenguaje escrito como “friendly use device”, que suena poco científica y académica, que se puede cambiar por “a user-friendly first-aid device”. Recomendaría la consistencia terminológica en algunas palabras como “Stop Nosebleeds” y “stop-nosebleed device” (idealmente en cursiva si es nombre comercial: Stop Nosebleeds), mejorar conectores ("likewise", "collectively", "for example"), mejorar la estructura del párrafo final ya que, el párrafo de cierre es largo, además de que se puede dividir en dos, mostrando las limitaciones del estudio y recomendaciones para futuras investigaciones. Conclusiones: Se repite dos veces seguidas "results"; ojo con la coherencia y redacción en los párrafos, mejorar los conectores igualmente, cerrar el escrito o el párrafo final enfatizando la relevancia del estudio y sus aportes. Bibliografía: Tiene más del 70% de fuentes científicas pasados los 5 años, están escritas en forma de Vancouver y normas APA, no hay coherencia en el escrito de cada una (si se escribe el Doi son a todas las fuentes no a medias).

Reviewer #5: Abstract: Appropriate and crysp

Introduction: Appropriate and satisfactoryReview

Materials & Methods: Research techniques and methodology used are appropriate

with standard statistical design application. Planning and conduct of experiments are

appropriate as per the topic chosen.

Results & Appendices: The result have been appropriately presented in an elaborate

and systemic manner, precisely interpreted and properly discussed with appropriate

discussion.

Discussion: Discussion is critical with valid justification addressing all the objectives

mentioned

Summary: Appropriate and satisfactory

**Do you want your identity to be public for this peer review?** For information about this choice, including consent withdrawal, please see our Privacy Policy

Reviewer #1: **Yes: ** Qaes T. Al-Obaidi

Reviewer #2: No

Reviewer #4: No

Reviewer #5: **Yes: ** Dr R H Bhatt

---

## [Editor Report · Acceptance letter]

PONE-D-24-52201R1

PLOS ONE

Dear Dr. Abu-Shaheen,

I'm pleased to inform you that your manuscript has been deemed suitable for publication in PLOS ONE. Congratulations! Your manuscript is now being handed over to our production team.

Kind regards,

on behalf of

Dr. Mehmet Baysal

Academic Editor

PLOS ONE